# Epithelial development of the urinary collecting system in the human embryo

**Marie Ange Saizonou[1], Haruka Kitazawa[1], Toru Kanahashi[1], Shigehito Yamada[1,2], Tetsuya Takakuwa [1] ***

1 Human Health Science, Kyoto University Graduate School of Medicine, Kyoto, Japan, 2 Congenital Anomaly Research Center, Kyoto University Graduate School of Medicine, Kyoto, Japan

* tez@hs.med.kyoto-u.ac.jp

**Data Availability Statement:** Yes - all data are fully available without restriction All relevant data are within the manuscript and its Supporting information files.

## Abstract

The urinary collecting system (UCS) consists of organized ducts that collect urine from the nephrons and transport it to the ureter and bladder. Understanding the histogenesis of the UCS is critical. Thirty human embryos between the Carnegie stages (CS) 18 and 23 were selected from the Congenital Anomaly Research Center, Kyoto, Japan. Epithelia of the UCS, ureter, and bladder of each sample were randomly selected. Histological findings of the epithelia were analyzed according to the following criteria: type of epithelium, presence or absence of glycogen, percentage of migrated nuclei, percentage of cells in mitosis, and the surrounding mesenchyme. A thickened epithelium lining a narrow luminal cavity was observed in the pre-expanded pelvic specimens at CS18-CS23. At CS23, after pelvic expansion, the UCS showed a thin epithelium with a large luminal cavity mainly located on the early branches, whereas the epithelium covering the subsequent branches had medium thickness. Histological characteristics differed depending on the UCS part and sample stage. The degree of differentiation was evaluated, revealing that in CS18-CS23 pre-expanded pelvis specimens, the undifferentiated epithelium was found in the zeroth to third/fifth generation, whereas at CS23, after pelvic expansion, a differentiated epithelium covered the UCS zeroth to seventh generation. In a comparison of the urothelial epithelium between the UCS, ureter, and bladder, we found that urinary tract differentiation may be initiated in the bladder, followed by the ureter, UCS zeroth to seventh generations, and finally, UCS eighth to end generations. An understanding of the histogenesis of embryonic stage UCS can aid in the clinical management of congenital urinary tract defects and other diseases.

## Introduction

The uroepithelium or urothelium is an epithelial tissue that lines the distal portion of the urinary tract, including the renal pelvis, ureters, bladder, and upper urethra and is composed of apical, intermediate, and basal cell layers [1]. Functionally, it forms a distensible barrier that accommodates significant changes in urine volume while preventing unregulated exchange of substances between the urine and blood supply. In addition to its role as a barrier, the uroepithelium can modulate the movement of ions, solutes, and water across the epithelial tissue [2]. The uroepithelium is, thus, a dynamic tissue that responds to changes in its local environment and can relay this information to other tissues that comprise organs [1].

**Funding:** SPS KAKENHI (grant number JP 21K07772, JP 23K14976). The funders had no role in study design, data collection and analysis, decision to publish, or preparation of the manuscript.

**Competing interests:** The authors have declared that no competing interests exist.

The bladder epithelium is derived from the endoderm of the vesical part of the urogenital sinus. The ureter originates from the ureteric bud, a protrusion of the mesonephric duct, during the development of the genitourinary system. Several studies have described urothelial differentiation in the bladder and ureter. Previous studies have reported that the urothelium is pseudo-stratified. However, later studies showed that it is stratified and that cytoplasmic processes are observed, although rarely, in the intermediate cell layers but not in the basal cell layers [3]. Wesson [4] demonstrated that a single layer of low-cuboidal to high-columnar epithelium with prominent vesicular features lines the entire vesicoureteric anlage. Felix [5] stated that this single layer over the openings of the ureters develops into two or three layers by the end of the 6th week and four to five cells thick by the 9th week.

The urinary collecting system (UCS) is a significant component of the metanephros, and its developmental process is initiated from a simple epithelial tube [6]. The UCS arises from the iterative branching of this epithelial tube, resulting in a series of elaborate ducts that collect urine from all the nephrons and transfer it to the bladder via the urothelial duct. Oliver [7] and Al-Awqati et al. [8] described the UCS as a highly organized and regulated structure of ducts formed by bifid branching during the embryonic and fetal periods, where bifurcation occurs approximately 15 times. In stage CS23, the proximal UCS is remodeled into the renal pelvis and calyx, from which the collecting ducts are distributed [9–11]. This phenomenon divides late embryos into two categories: pre-expanded and expanded pelvis groups [12]. Several factors, including the development of nascent nephrons, their connection to the UCS, and the initiation of urinary excretion, may contribute to proximal UCS expansion [12, 13]. However, the mechanisms underlying proximal UCS expansion remain unclear. Potter et al. demonstrated an epithelium composed of tall columnar cells with large oval nuclei, surrounded by connective tissue, in early embryos [14]. In late embryos, the expansion of the first generation of branches presented a thin epithelium. Our group has provided a brief overview of histogenesis, mainly focusing on UCS morphogenesis and branching development [12]. Pelvic expansion could affect UCS histology and vice versa. Precise differentiation of UCS epithelium according to the number of generations is lacking. Recent studies using experimental mouse models have provided a histological atlas of the developing mouse urogenital system describing organogenesis and morphogenesis. It is unclear whether these observations can be applied to the human UCS, as the early branches of the UCS in late mouse embryos appear to be different from those in humans [15].

Although the urinary tract (including the ureter and bladder) is uniformly lined by the urothelium, the process of histological differentiation during the embryonic period is primarily confined to the bladder [2, 3, 16] and urinary tract [6], with detailed descriptions lacking for the UCS [12]. Because urine secretion initiates during the late embryonic period [17], clarifying the features and timeline of the histological differentiation of the urothelium region by region is essential. Clinically, this could yield data to ascertain the locations of congenital urinary tract defects, potentially facilitating the initiation of a clinical retrospective study on probable causes. Herein, we aimed to demonstrate the differentiation of the UCS epithelium in the human metanephros during the human embryonic period and to evaluate its degree of differentiation compared to that of the ureter and bladder epithelia.

## Materials and methods

### Ethics approval

The ethics committees of Kyoto University Faculty and Graduate School of Medicine, Japan, approved the use of human embryonic and fetal specimens for this study (approval number: R0316).

## Human embryonic specimens

At the Congenital Anomaly Research Center, Kyoto University Graduate School of Medicine, Japan, are stored in nearly 45,000 human specimens of embryos and fetuses, constituting the Kyoto Collection [18]. Most specimens were obtained when the pregnancy was terminated during the first trimester under the Maternity Protection Law of Japan. Samples were collected from 1961 to 1971, according to the regulations pertaining to each period (Table 1). For example, written informed consent was not required from parents back then; instead, verbal informed consent was sufficient to deposit the specimens, which was documented in medical records. All samples were anonymized and de-identified. Data were accessed for research purposes from Apr 1, 2021, to Mar 1, 2023. Approximately 20% of the specimens were undamaged and well-preserved. Embryos were measured, examined, and staged according to the criteria described by O'Rahilly and Müller [19]. Whole embryonic samples were fixed with 10% formalin, embedded in paraffin, and serially sectioned to a thickness of 10 μm. These samples were stained with hematoxylin and eosin (HE), and the specimens were preserved.

In this study, serial tissue sections were used from embryo specimens between CS18 and CS23 belonging to the Kyoto Collection ranging from 12.0 to 28.7 mm crown-rump length (CRL) (n = 30; five samples per stage) (Table 1).

## Digitalization of histological sections and metanephros 3-D reconstruction

Histological sections of the metanephros were digitalized, and three-dimensional (3D) reconstructions were generated as described previously [12]. Briefly, serial transverse sections (thickness, 10 μm) of whole embryos were digitalized using an Olympus virtual slide system (VS120-S5-J, Olympus Corp., Tokyo, Japan) for histological observations and 3D reconstructions. Sequential two-dimensional (2D) images at 25× magnification were digitally cropped around the metanephros. The metanephros, including the UCS, was segmented into serial digital sections. 3D images and the centerline of the UCS were computationally reconstructed and analyzed using Amira v. 5.5.0 (Visage Imaging GmbH, Berlin, Germany).

## Selection of histological sections for analyses of UCS epithelium as per the generation number, urothelial duct, and bladder

Using the reconstructed UCS tree, sections that included branches suitable for histological observations were selected. These sections included the transverse section of the branches in the middle, where the generation numbers could be identified. The selected images were cropped at 40× magnification and used for histological observation and analysis (Fig 1). Four transverse sections per generation of each sample were selected for further observation (Fig 1A and 1B).

**Urothelial duct.**   Representative sections of the urothelial duct located at the ureteric-pelvic junction (UPJ) and bladder (distal section) were selected (Fig 1B). The selected images were cropped at 40× magnification and used for histological observation and analysis. Two transverse sections for each sample were selected for further observation.

**Bladder.**   For each specimen, whole sections of the bladder, including anatomical landmarks, were identified. Suitable sections were cropped, magnified 40× on the bladder neck and dome regions, and used for histological observation and analysis (Fig 1B). Two sections from two samples each per stage were selected for further observation.

**Table 1. Samples used in the present study.**

| CS | ID | Collecting date | CRL | Orientation | Serial sections | | Urinary collecting branches | |
|---|---|---|---|---|---|---|---|---|
| | | | | | Total number | Kidney Range | Maximum number of generation | Total branches |
| **18** | 24992 | 14.02.1970 | 12 | Cr | 39 | 30–33 | 4 | 11 |
| | 28129 | 10.04.1971 | 13 | Cr | 50 | 37–43 | 4 | 15 |
| | 10309 | 1.06.1966 | 13.7 | Sa | 31 | 11–24 | 5 | 17 |
| | 15391 | 19.07.1967 | 14.2 | Lo | 26 | 12–21 | 2 | 5 |
| | 3901 | 17.08.1965 | 14.7 | Lo | 43 | 22–27 | 6 | 25 |
| **19** | 7168 | 02.02.1966 | 13.3 | Lo | 96 | 16–21 | 7 | 35 |
| | 16696 | 25.11.1967 | 13.7 | Cr | 55 | 47–49 | 7 | 27 |
| | 8389 | 21.06.1966 | 16.3 | Cr | 83 | 25–35 | 6 | 39 |
| | 3002 | 05.06.1965 | 18.4 | Cr | 87 | 70–75 | 7 | 37 |
| | 923 | 29.08.1963 | 17.5 | Sa | 39 | 13–30 | 7 | 58 |
| **20** | 2006 | 05.04.1965 | 15.9 | Cr | 62 | 58–59 | 7 | 59 |
| | 7271 | 19.02.1966 | 16.9 | Cr | 91 | 75–79 | 7 | 69 |
| | 4330 | 04.07.1965 | 18.6 | Cr | 87 | 75–78 | 7 | 90 |
| | 1580 | 08.01.1965 | 18.8 | Lo | 111 | 46–55 | 8 | 75 |
| | 567 | 12.04.1962 | 20.8 | Sa | 200 | 22–44 | 8 | 101 |
| **21** | 12155 | 27.10.1966 | 17.9 | Cr | 80 | 41–52 | 9 | 141 |
| | 16393 | 07.04.1967 | 18.7 | Cr | 60 | 46–51 | 8 | 141 |
| | 41 | 09.06.1961 | 19 | Cr | 98 | 44–57 | 9 | 102 |
| | 2021 | 30.03.1965 | 21.4 | Sa | 120 | 43–94 | 9 | 120 |
| | 2314 | 31.03.1965 | 22.6 | Cr | 142 | 69–93 | 9 | 160 |
| **22** | 8825 | 16.05.1966 | 21.3 | Cr | 115 | 65–83 | 10 | 182 |
| | 5685 | 17.11.1965 | 21.7 | Cr | 185 | 123–142 | 9 | 147 |
| | 9305 | 12.05.1966 | 22 | Cr | 96 | 79–86 | 11 | 286 |
| | 10444 | 12.08.1966 | 22.5 | Cr | 142 | 93–115 | 9 | 250 |
| | 5214 | 24.09.1965 | 23.4 | Cr | 201 | 103–135 | 11 | 333 |
| **23** | 3104 | 27.07.1965 | 25.2 | Lo | 137 | 96–113 | 9 | 216 |
| | 9005 | 23.04.1966 | 26 | Sa | 197 | 60–135 | 11 | 416 |
| | 4381 | 27.08.1965 | 26.3 | Sa | 198 | 26–104 | 11 | 480 |
| | 9026 | 03.06.1966 | 25.7 | Sa | 209 | 51–132 | 12 | 501 |
| | 12481 | 31.08.1966 | 28.7 | Sa | 205 | 104–190 | 12 | 397 |

CS, Carnegie stage; collecting date, date when the samples were first collected; CRL, crown-rump length; Cr, cross-section; Sa, sagittal section; Lo, longitudinal section; kidney range, serial sections where kidney was observed.

### Evaluation of the UCS epithelium, urinary duct, and bladder

**Histological findings.** Histological findings of the UCS and transition side of the ureter were evaluated based on the criteria of a previous study describing the choroid plexus, with minor modifications [20, 21]. Characteristics of the UCS were selected based on these criteria, adapted, and reorganized according to our histological findings and the specificity of the UCS. These include 1) the type of epithelium (pseudostratified epithelium E1, simple cuboidal epithelium E2), 2) perinuclear glycogen in the cytoplasm (presence or absence), 3) percentage of migrated nuclei, 4) percentage of nuclei in mitosis, and 5) mesenchymal tissues surrounding the epithelium (loose connective tissue [CT], metanephric blastema [MB]) (Table 2).

Histological evaluation of the differentiation is not known for the human urothelium and is limited to another type of epithelium except for that of the choroid plexus [20, 21]. The

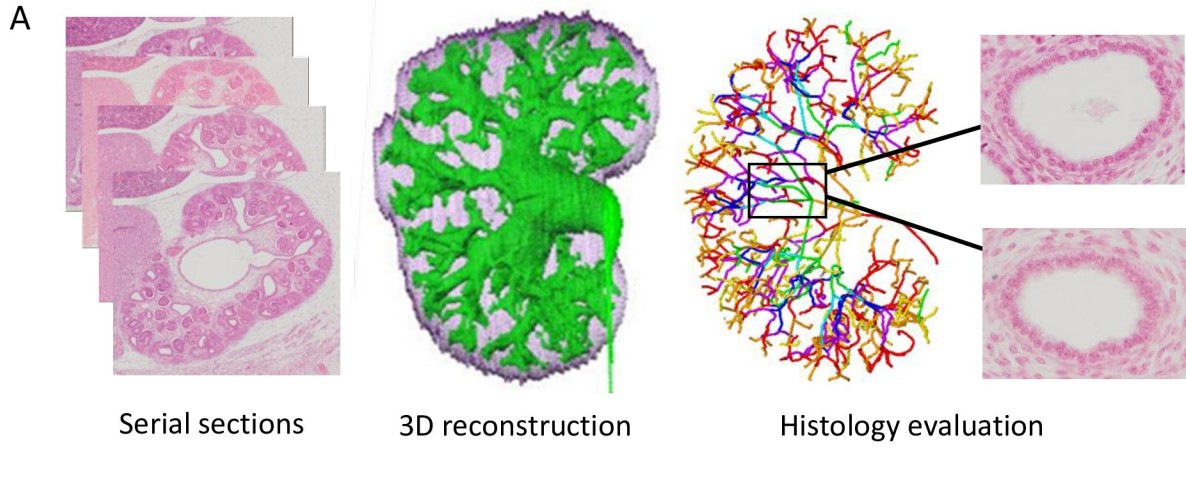

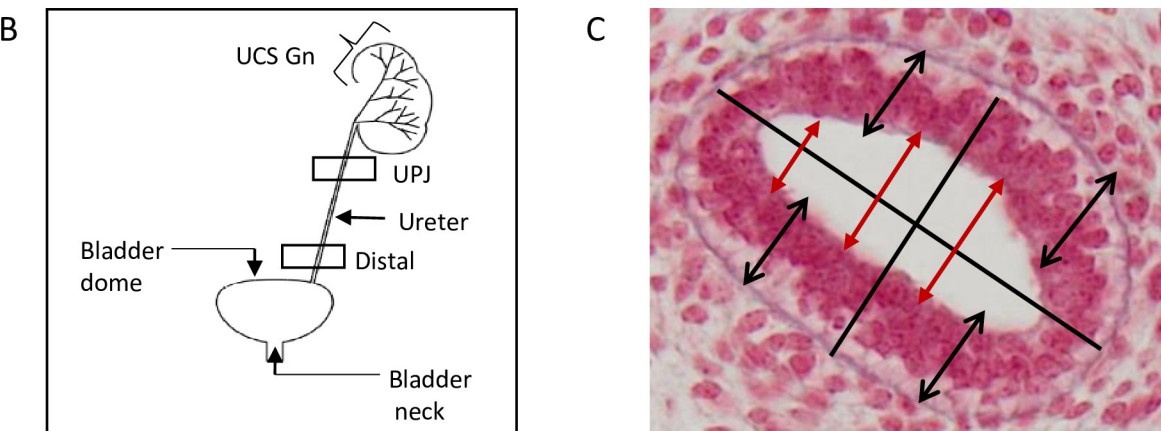

**Fig 1. Methods used in the present study.** (A) Selection of histological sections for the analysis of the urinary collecting system (UCS) epithelium according to the generation number. Three-dimensional reconstruction of metanephros and UCS centerline tree was used for histological evaluation. Representative generation branches were selected. Matched epithelium sections were cropped at 40× magnification. Illustrative data were obtained from a representative metanephros sample (CS23: ID 9026). (B) Illustration indicating the sections observed on the urinary tract. UPJ, ureteric pelvis junction; UCS Gn, urinary collecting system generation number. (C) Measurements of epithelium height and lumen diameter. The height of the epithelium and the size of the lumen are determined by averaging the lengths measured at four positions (indicated by black bidirectional arrows) and three positions (indicated by red bidirectional arrows) on the UCS, respectively.

choroid plexus is a special organ that produces cerebrospinal fluid and is responsible for an important biological barrier system, forming an interface between the blood and the cerebrospinal fluid. Considering the urothelium's several functions described above, some physiological similarities in both structures appear evident. Therefore, it seemed interesting to use a similar differentiation criterion to describe epithelial tissue on the urinary tract whilst including its distinctive features.

The differentiation score in the UCS was calculated by the sum of the scores from each criterion.

The distribution of nuclei within the epithelium, observed in both apical-central and apical-central-basal regions—a characteristic akin to that of apically migrated nuclei—served to enrich the characterization of the ureteral epithelium.

**Table 2. Histological characteristics and differentiation scores of the UCS.**

| Structure | State | Score 0/0.5 | Score 1 |
|---|---|---|---|
| Epithelium | Type | Pseudostratified epithelium (E1)<br>Epithelium covered by a disposition of crowded oval nuclei located in various positions | Simple cuboidal epithelium (E2)<br>Epithelium lined by simple cuboidal cells |
| | Glycogen | Absence G (-)<br>Perinuclear substance inside the cytoplasm absent | Presence G (+)<br>Perinuclear substance inside the cytoplasm present |
| Nuclei | Apical migrated nuclei | High (>40%)<br>Intermediate ([15%–40%])<br>Percentage of apical migrated nuclei greater than 40% (Score 0)<br>Percentage of apical migrated nuclei ranged between [15%–40%] (Score 0.5) | Low (< 15%)<br>Percentage of apical migrated nuclei less than 15% |
| | Percentage of nuclei in mitosis | High (> 2%)<br>Percentage of nuclei in mitosis greater than 2% | Low (< 2%)<br>Percentage of nuclei in mitosis less than 2% |
| Mesenchymal tissue surrounding the epithelium | Connective tissue/ Metanephric blastema | Metanephric blastema (MB)<br>Mesenchyme surrounding UCS: metanephric blastema | A loose connective tissue (CT)<br>Loose connective tissue surrounding UCS |

The bladder epithelium was observed at the neck and dome based on previous criteria [20, 21], according to the following histological findings: 1) type of epithelium (single layer epithelium [SL]), bi-layered epithelium [BL], multilayered epithelium [ML]), 2) perinuclear glycogen in the cytoplasm (present or absent), 3) presence of nuclei in mitosis, and 4) mesenchymal tissues surrounding the epithelium.

**Measurements.** The epithelial height and lumen diameter were measured in two selected sections per generation of branches in each sample. Two perpendicular lines, that is, the long and short axes, were determined. The epithelium height was measured parallel to the short axis at four positions. The epithelium height was calculated as the mean of the four lengths (Fig 1C). The lumen diameter was measured parallel to the short axis at three points and was calculated as the mean of the three lengths.

## Results

### 3D reconstruction of the UCS tree

3D reconstruction of the UCS tree revealed growth during the embryonic period (Fig 2), with an increase in the number of UCS generations and total branches. The maximum generation number (and ranges) of UCS end branching at each stage were as follows: CS18, 4.2 [2–6]; CS19, 6.6 [6–7]; CS20, 7.2 [6–8]; CS21, 8.8 [8–9]; CS22, 9.4 [9–10]; CS23P, 10.6 [10–11]; and CS23E, 11.0 [10–12]. The medians (and range) of the total number of UCS branches by the sixth to end generations (shown in orange in Fig 2) were as follows: CS18, 0.4 [0–2]; CS19, 7 [2–17]; CS20, 21.8 [10–47]; CS21, 63.4 [43–91]; CS22, 177.4 [87–268]; CS23P, 326.0 [140–420], and CS23E, 392.0 [334–450] (See also Table 1).

### Epithelium height according to growth

The epithelial height between CS18 and CS21 was approximately distributed between 20–25 μm, accompanied by an increase in generation number (Fig 3A). It markedly increased in the first to second generations at CS22, reached the local maximum at the second generation, and gradually decreased with an increase in the generation number. A similar trend was

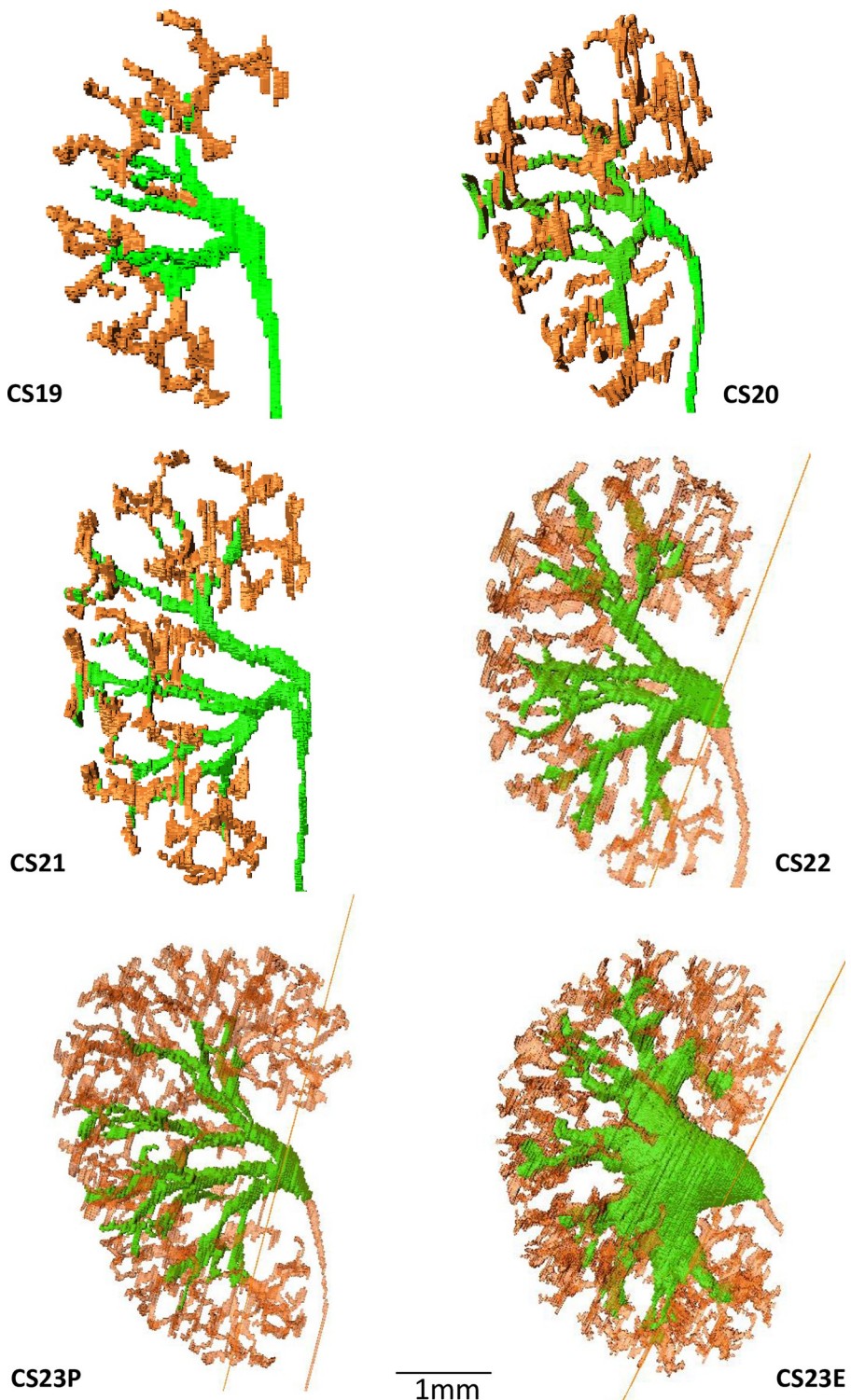

**Fig 2. Three-dimensional reconstruction of metanephros at stages CS18-CS23.** Green, zeroth to the fifth generation; orange, sixth to the end generation. The specimens at CS23 are divided into two categories: the preexpanded pelvis group (CS23P) and the expanded pelvis group (CS23E), leading to an enlargement of the pelvis. Data was obtained from representative specimens at each Carnegie stage (CS19: ID923, CS20: ID567; CS21: ID2021; CS22: ID5685; CS23: ID9026).

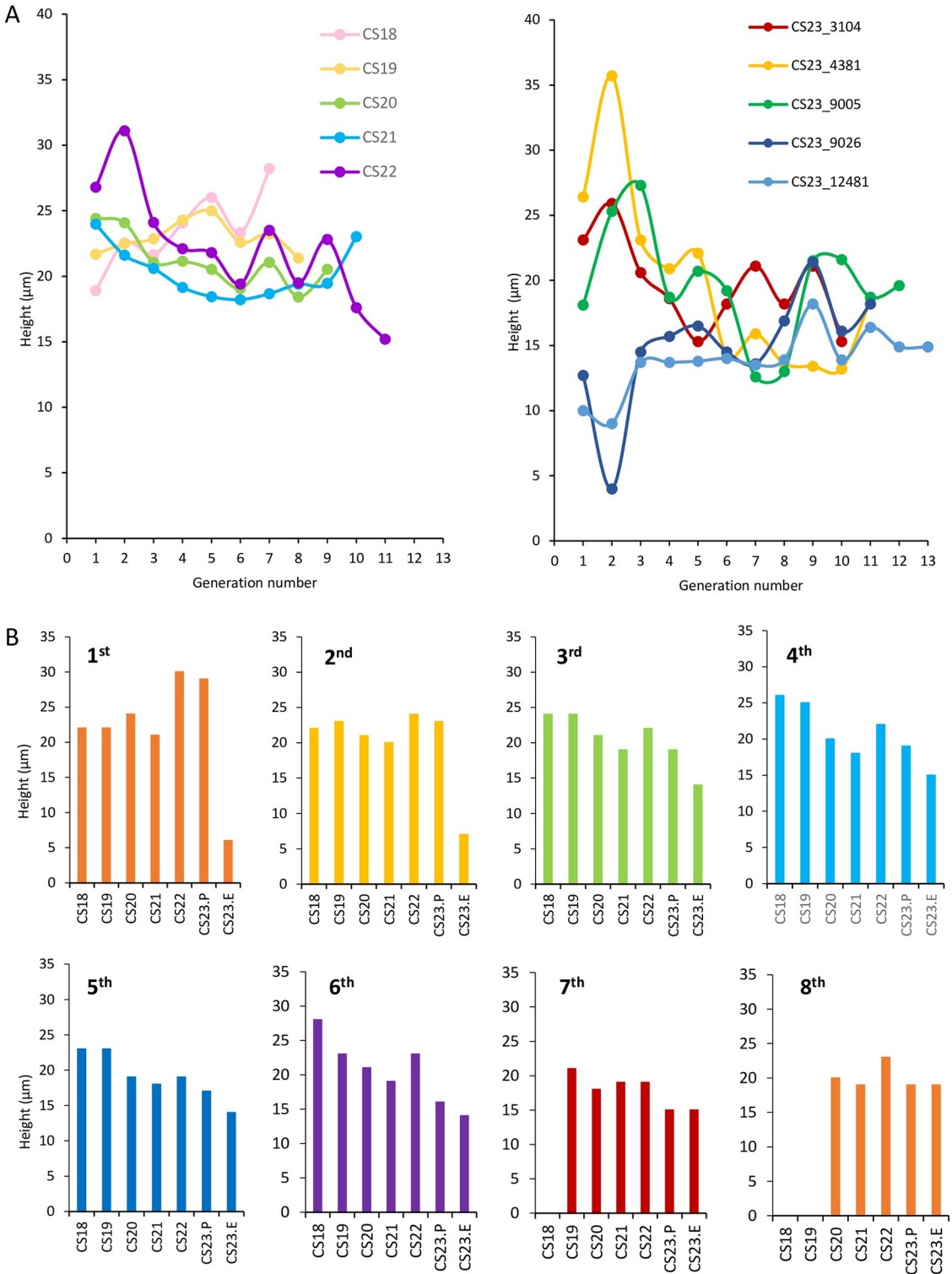

**Fig 3. Epithelium height in the UCS in relation to generation number.** (A) Change in the epithelium height according to the generation number. The mean thickness vs. generation number between CS18 and CS22 is indicated on the left, and the data value at each sample at CS23P and CS23E is indicated on the right. (B) Comparison of the epithelium heights between CS18 and CS23E by generation number. CS, Carnegie stage; CS23.P, Carnegie stage 23 pre-expanded pelvis; CS23.E, Carnegie stage 23 expanded. The measurement data was provided in S1 Data.

observed at CS23P. The epithelial height markedly decreased from the first to the seventh generations at CS23E, mainly from the first to the fourth generations, with a local minimum observed in the second generation.

The epithelial heights of each generation were compared. In the first generation, the epithelial height showed a small local maximum at CS22 and CS23P. In the second and peripheral generations, epithelial height decreased as the stage increased, whereas the local maximum at CS22 became inconspicuous (Fig 3B).

## Lumen size according to growth

The average cavity size between CS18 and CS21 was almost similar (10–20 μm), slightly decreasing as the generation number increased. The lumen size increased from the first to fourth generations in the two samples of CS23P (Fig 4A). Lumen size increased with each generation of CS23E. The increase in lumen size was more prominent at CS23E than at CS18 -CS23P (Fig 4B).

## Comparison of epithelium height and lumen size by generation number

In the previous stages, between CS18 and CS23P, epithelial thickness and lumen size did not correlate, and lumen size was almost constant, regardless of epithelial thickness. At CS23E, the UCS showed a thin epithelium with a large luminal cavity, and its distribution differed from that in the previous stages (Fig 5). The trend was similar in all scatter plots for each generation number.

## Histological evaluation

**Epithelium type.** The UCS epithelium between CS18 and CS23P consisted of crowded oval nuclei located at various positions and a pseudostratified epithelium (E1) until their respective end branches. The epithelium on CS23E was lined with a superficial layer of cuboidal cells (E2), well-aligned around the lumen from the zeroth to the seventh generation. In contrast, the seventh-end generation exhibited characteristics similar to those of E1.

**Glycogen.** The perinuclear transparent substance glycogen, which appeared to repel the nuclei towards the apical side, was observed from the zeroth to second generations at CS18, from the zeroth to the fifth generations at CS19 and CS23P, and from the zeroth to the sixth generation at CS23E. Glycogen was not observed in the peripheral epithelium.

**Apical migrated nuclei and nucleus position.** At CS18-CS23P, nuclei were distributed mainly at the apical-central-basal position in the epithelium in the zeroth to end generations. The percentage of apical migrated nuclei in each epithelial layer was high through generations, ranging from 40% to 45%. At CS23E, nuclei were distributed mainly at the apical-central position in the epithelium from the zeroth to seventh generations and at the apical-central-basal position between the eighth to twelfth generations. The percentage of apical nuclei was lower in the zeroth to seventh generations (3%–15%), increasing slightly in the eighth to end generations (12%–28%), and remained lower than in embryos between CS18 and CS23P.

**Percentage of nuclei in mitosis.** In most regions of CS18 and CS20 and the peripheral regions of CS19 and CS23E, mitosis was observed in more than 2% of the epithelial cells. Mitosis was observed in less than 2% epithelia, mainly in the proximal part of the UCS, with several exceptions between CS21 and CS23E, namely, zeroth to fourth generations at CS21, zeroth to fifth generations except for first and second generation at CS22, and zeroth to sixth generations at CS23P and CS23E.

**Mesenchymal tissue surrounding the epithelium.** Loose connective tissue surrounded the UCS epithelium in the zeroth to second generations at CS18, fourth generation at CS19

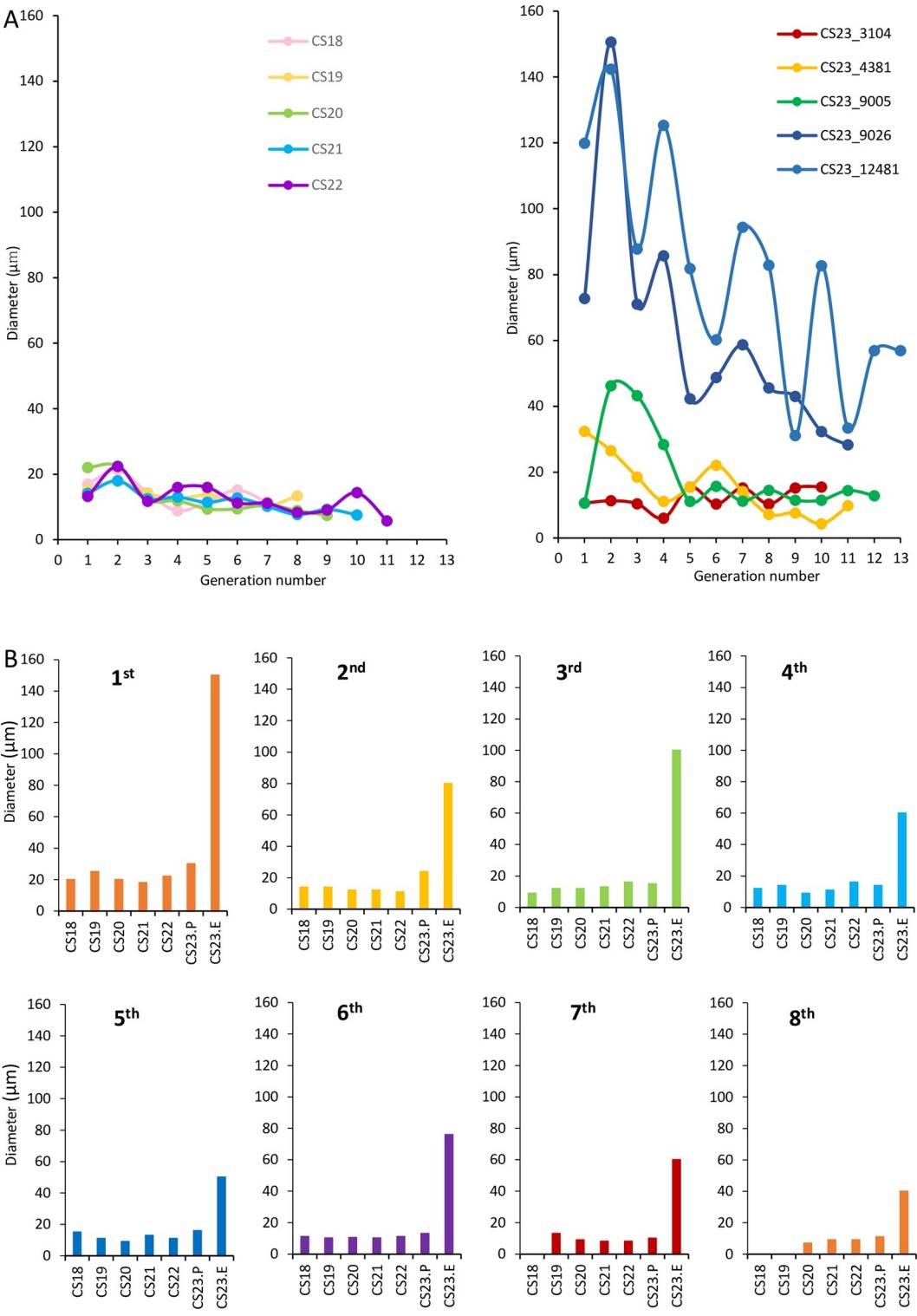

**Fig 4. Lumen size in the UCS according to generation number.** (A) Change in the lumen size according to the generation number. The mean lumen size vs. generation number between CS18 and CS22 is indicated on the left, and the data values for each sample at CS23P and CS23E are indicated on the right. (B) Comparison of the epithelium lumen sizes between CS18 and CS23E by each generation. CS, Carnegie stage; CS23P, Carnegie stage 23 pre-expanded pelvis; CS23E, Carnegie stage 23 expanded. The measurement data was provided in S2 Data.

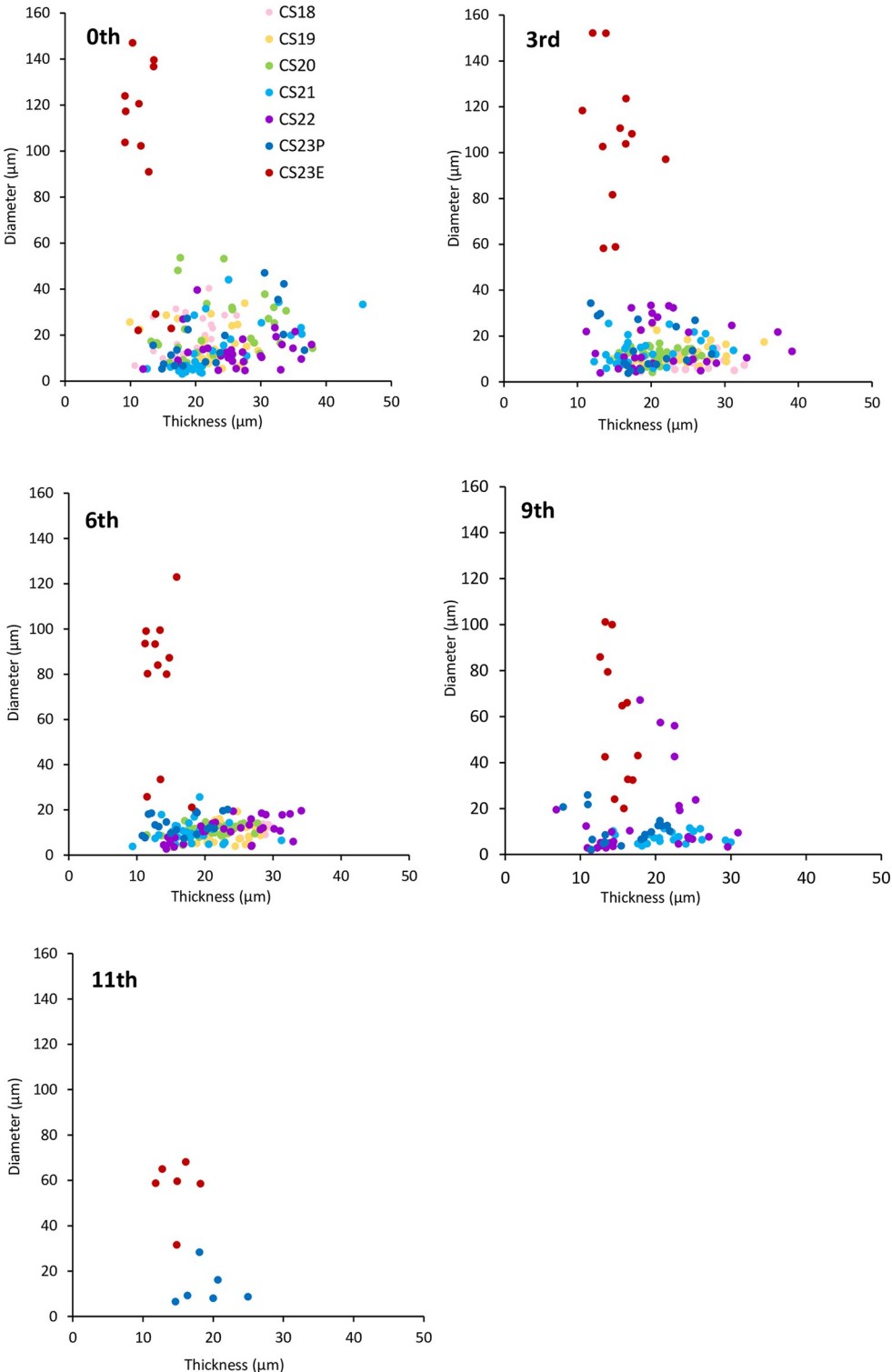

**Fig 5. Scatter plots showing the relationship between epithelium height and lumen size of the UCS during CS18-CS23E.** Scatter plots are provided for five representative generation numbers: 0th, 3rd, 6th, 9th, and 11th. The measurement data was provided in S3 Data.

and CS20, zeroth to sixth generations at CS21, and zeroth to seventh generations at CS22, CS23P, and CS23E. The metanephric blastema surrounded the subsequent peripheral branches.

## Differentiation score

The differentiation score of the UCS epithelium in each generation was calculated as the sum of the differentiation degrees in the five criteria mentioned earlier (Table 2). The histological features and differentiation in the proximal UCS region differed from those in the peripheral regions, depending on the specimen's embryonic stage (Fig 6A). The differentiation score increased according to the increase in CS and was higher in the proximal region than in the peripheral region. The score was high in the zeroth to seventh generations for CS23E. Representative histological pictures corresponding to the highlighted rectangle in the panel in Fig 6B were presented in Fig 7.

## Ureter

Glycogen-rich epithelium (G(+)) and surrounding loose connective tissue (CT) were observed at all stages, at both the junctional and distal sides (Fig 8A). Nuclei were distributed mainly at the apical-central-basal position in the epithelium between CS18 and CS22 and at the apical-central position at CS23P and CS23E at both the junctional and distal sides. Pseudostratified epithelium (E1) was observed between CS18 and CS23P at both the junctional and distal sides. At CS23E, a single cuboidal epithelium (E2) was observed at the junction side (close to the pelvis). Single cuboidal and bilayered epithelium was observed on the distal side (close to the bladder). The representative histological pictures are presented in Fig 8B.

## Bladder

Glycogen-rich epithelial wall (G(+)) and surrounding loose CT were observed at all stages in both the dome and neck of the bladder (Fig 9A). The bladder epithelium appeared to be different from that observed in the UCS, with progressive differentiation. The epithelium at CS18-CS19 was lined by a glycogen-rich, superficial layer of cells at the dome of the bladder and by multilayer cells at the neck of the bladder. At the bladder dome, the number of layers increased with growth; a BL was observed from CS19 onwards, and a tri-layered epithelium was observed from CS21 onwards. At all stages, the neck of the bladder was covered with ML. The representative histological pictures are presented in Fig 9B.

## Discussion

The urothelium lines the urinary tract, from the metanephros to the urethra. Previous studies on urothelial histology mainly focused on the bladder [5, 16] and urethra [4]. Newman and Antonakopoulos [16] reported that the proximal bladder epithelial layers progressively increase from a monolayer to a bilayer and then to a trilayer epithelium. Wesson [4] reported that a single layer of low-cuboidal to high-columnar cells lines the ureteral structure. These classical studies shed light on the general histological differentiation of the urothelium.

Few studies have focused on urothelial differentiation in the UCS. Potter [14] focused mainly on kidney morphogenesis and the spatial distribution of UCS branches and nascent nephrons. Limited observations have been made of the UCS epithelium, namely the pseudostratified epithelium, which suggests the presence of a multilayered epithelium. Lindström et al. reported on the 3D morphogenesis of the kidney by comparing human and mouse kidneys [15]. In UCS histology, a pseudostratified ureteral epithelial branch tip was observed at stages

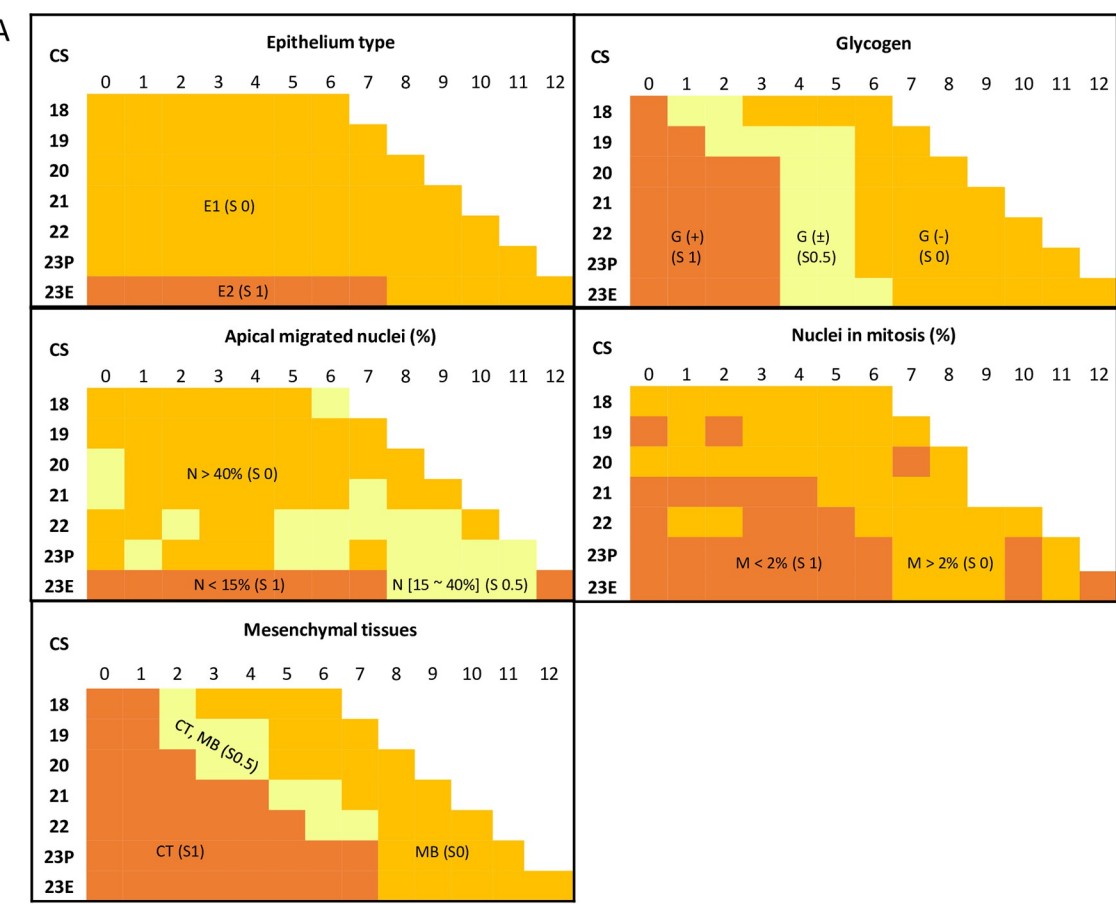

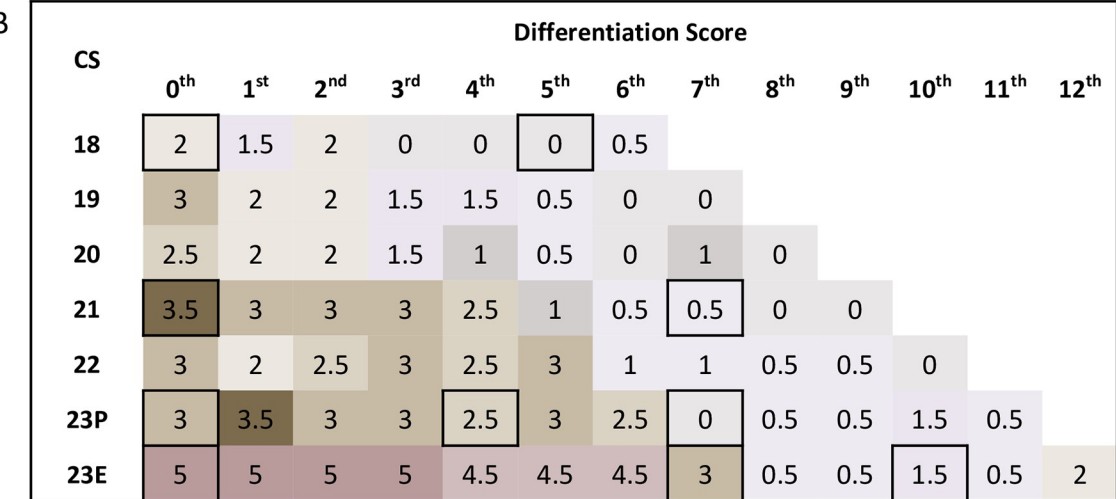

UCS/Generation number

**Fig 6. Histological examination according to the Carnegie stage and generation number.** (A) Each panel indicates the type of epithelium, presence of glycogen, percentage of migrated nuclei, percentage of cells in mitosis, and mesenchyme surrounding the epithelium. Each criterion is explained in the Materials and Methods section. (B) The differentiation score was obtained from the six histological examinations. Histological images used in Fig 7 are highlighted by rectangles.

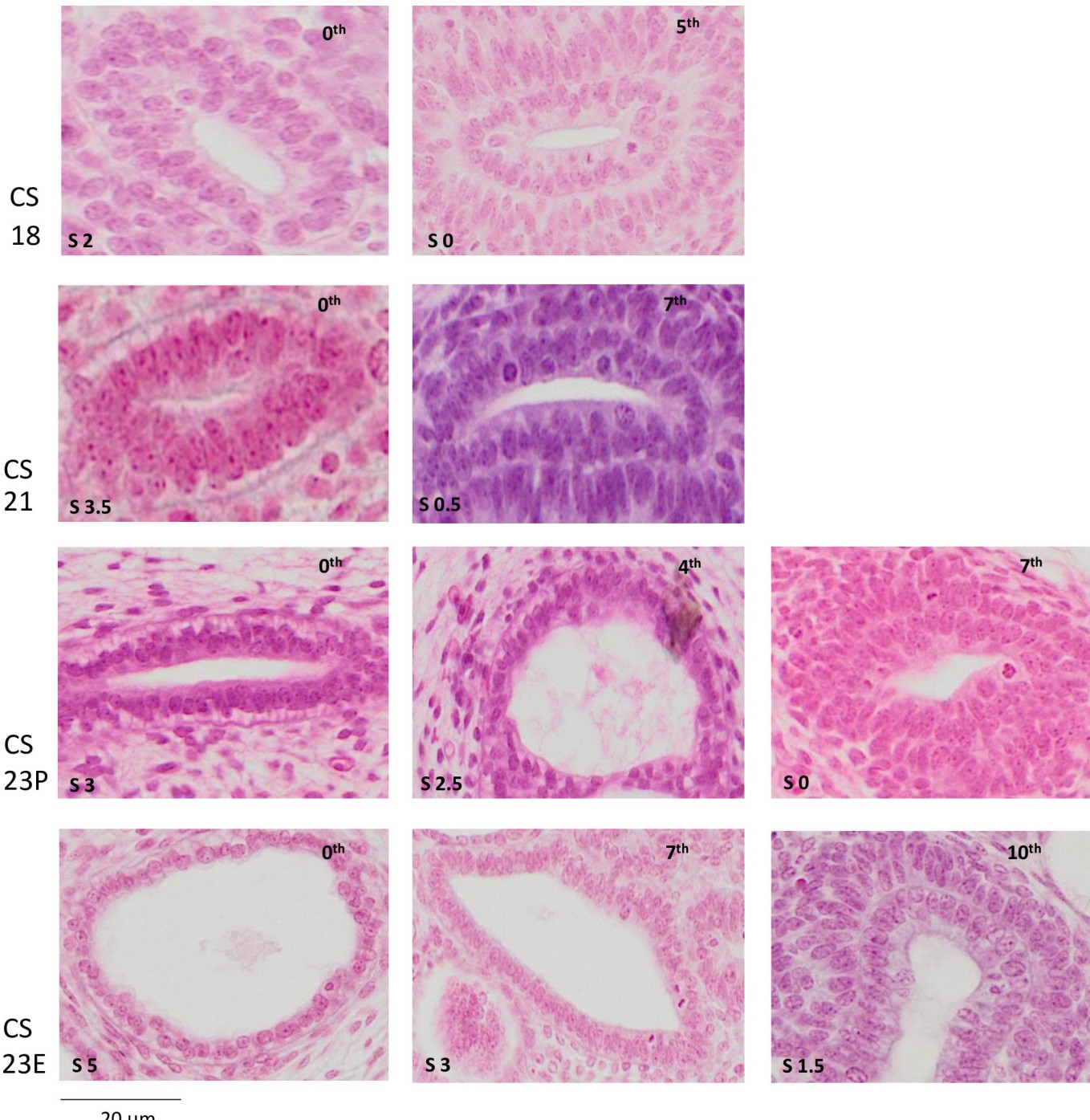

**Fig 7. Representative histological images depicting the UCS differentiation scores (S).** The generation number is mentioned in each image. Histological images presented here are highlighted by rectangles in Fig 6.

CS16-CS19. The present study was unique in that the histogenesis of the urothelium was highlighted in the whole urinary tract, consisting of the UCS, ureter, and bladder, stage-by-stage, during the embryonic period. Moreover, this enabled comparisons of differentiation within the bladder, urinary tract, and UCS, as well as among these three, on the differentiation

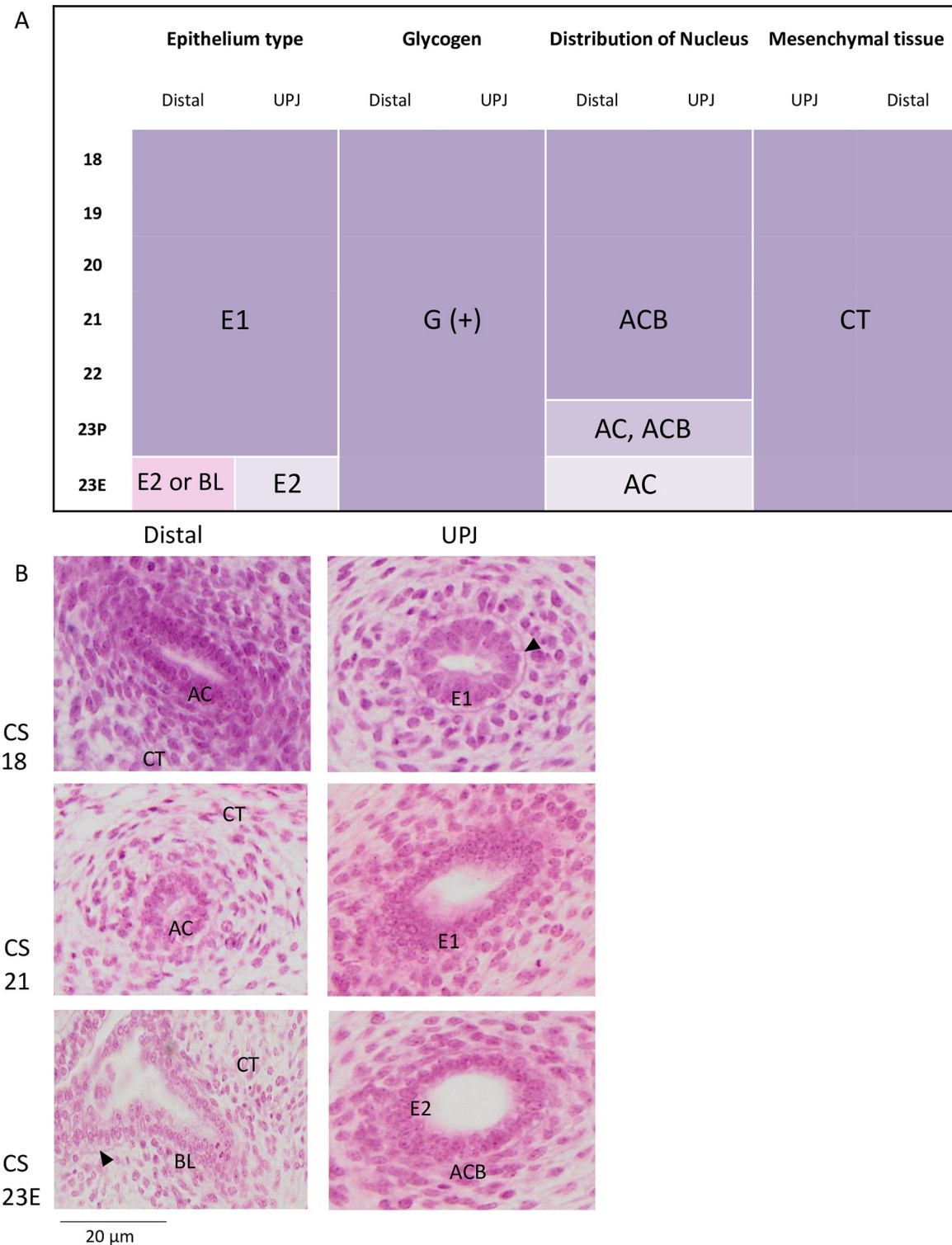

**Fig 8. Histological findings in the ureter.** (A) Panels indicate the epithelium type, presence of glycogen, nuclei position, and surrounding mesenchyme at the junction and the peripheral side. (B) Representative histological images of the ureter. E1, pseudostratified epithelium; E2, simple cuboidal epithelium; BL, bi-layered epithelium; G, glycogen; AC, apical-central; ACB, apical-central-basal; CT, connective tissue. Purple (in A) indicates the repartition of each feature by the Carnegie stage. Arrowheads indicate glycogen. Numbers represent Carnegie stage (CS); images shown on higher magnification (40×) ureter epithelium; scale bar = 20 μm. UPJ; ureteric pelvic junction, Distal; the distal side of the ureter.

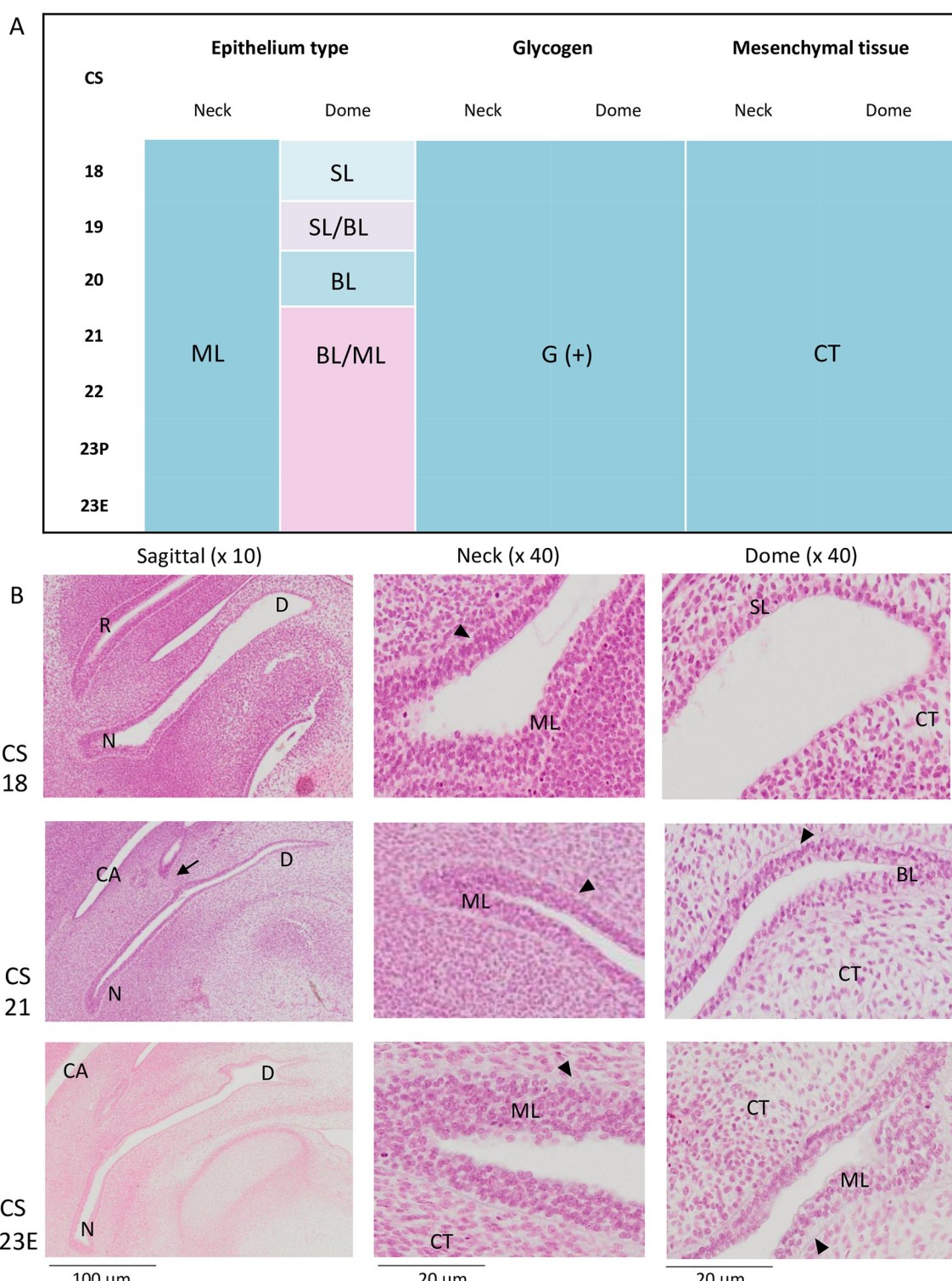

**Fig 9. Histological findings on the bladder.** (A) The panel indicates the epithelium type, presence of glycogen, and surrounding mesenchyme at the neck and dome of the bladder. SL, single layer; BL, bi-layers; ML, multilayers; G, glycogen; CT, connective tissue. (B) Representative histological images at CS18, CS21, and CS23E from the whole bladder (left), magnified neck (middle), and dome region (right). Arrows indicate the opening of the ureter. From left to right, respectively: lower magnification (10×) of sagittal section of whole bladder, scale bar = 100 μm; higher magnification (40×) showing neck and dome of bladder epithelium, scale bar = 20 μm. D, dome; N, neck; R, rectum; CA, abdominal cavity.

timeline. We have described the UCS differentiation in detail, and specifically in the epithelium by each generation number.

In the UCS, the differentiation score decreased from the proximal to peripheral parts (with increasing generation number), regardless of the Carnegie stage. This trend was more pronounced for CS23E. In the distal ureter, the epithelium resembled that of the bladder. Contrastingly, the epithelium mirrored that of the UCS in the transitional region at CS23E, suggesting that the epithelial differentiation was more pronounced in the distal region than in the transitional area at CS23E. By CS18, the bladder epithelium in the neck region had become multilayered, evolving from a monolayer into a multilayered structure at the dome between CS18 and CS23.

The bladder neck epithelium at CS18-CS19 was comparable to the UCS epithelium in the zeroth to seventh generations at CS23E. Our findings suggest that the initial differentiation of urothelial cells may occur in a retrograde manner across the urinary tract, beginning with the bladder, followed by the ureter, and subsequently, the UCS.

A pseudostratified epithelium was observed at CS18-22, as described in previous studies [14, 15]. We observed and quantified the position of nuclei in the pseudostratified epithelium as an indicator of differentiation. The nuclei frequently migrated to the apical side of the UCS epithelium at stages CS18-CS23P, affecting approximately 45% of the nuclei. This migration observed in our study could correspond to the partial interkinetic nuclear migration (INM) noted in the ureter [22] and several digestive organs [23].

Makiko hypothesized that INM is a general strategy for epithelial progenitor expansion, although discrepant hypotheses have also been proposed [24].

The present study is the first to investigate the histology of the UCS after pelvic expansion (CS23E). The UCS at CS23E was completely different from that at previous stages, as noted in the 3D reconstruction and histological findings. In the previous stages, the epithelial thickness and lumen size were not correlated, and the latter was primarily constant regardless of the former. At CS23E, the UCS showed a thin epithelium with a large luminal cavity.

Although the expansion in the proximal region was striking, size expansion was simultaneously observed up to the terminal branch at CS23E. In contrast, differentiation timing at the UCS periphery occurred last in comparison to the rest of the urinary tract. This expansion may indicate the initiation of the physiological function of the urinary tract and the excretion of urine. Potter hypothesized that urinary secretions contribute to the expansion of the pelvis and calyces [8]. Several studies have shown that glomerular filtrate may increase the UCS pressure, forcing it to expand rapidly. We detected nephrons from CS19 and found that the total number of nephrons, and those connected to the UCS, increased until CS23 [12]. The human metanephros contributes to amniotic fluid volume at CS23 or earlier [17]. Lindström et al. [15] indicated that glomerular filtration may occur at CS23.

A previous study on the differentiation of the choroid plexus demonstrated that the histological findings, including the type of epithelium, presence of glycogen, position of nuclei, percentage of cells in mitosis, and surrounding mesenchyme, changed during the embryonic period between CS18 and CS23 [24]. These changes correspond to Netsky's Stages I and II, the first two of the four stages of differentiation [20, 21]. In the present study, most of these criteria were used to estimate urothelial differentiation during the same embryonic period. This means that urothelial and columnar (secretory) epithelia show similar findings during initial differentiation with a similar timeline in the human embryonic period. The airway epithelium shows findings similar to those of UCS. In 7-week embryos, the airways are lined by a thick pseudostratified endodermal epithelium in a bed of abundant loose mesenchyme; at 8 weeks, the airways enlarge and are covered by a simple columnar ciliated epithelium with prominent vacuoles [25]. The squamous epithelium of the skin shows an undifferentiated monolayer of

epithelium with glycogen and various mitoses in seven-week embryos [26]. The present study suggests that the histological findings during the initial UCS differentiation are similar to those of other epithelia, including secretory and squamous epithelia with similar timelines.

Three issues need consideration in this study. First is the effect of long-term storage on the subjects. Second, while tissue sections facilitate detailed observation, our quantitative histological analysis, employing formalin- and paraffin-embedded materials, may have been susceptible to errors. These errors could arise from degeneration during specimen handling, shrinkage due to fixation, and shearing during sectioning. Third, four suitable transverse sections were selected for each generation per sample. Accurate transverse sections were sometimes difficult to obtain in younger generations (proximal part) because the number of branches was limited.

## Conclusion

This study investigated the development of the UCS, ureter, and bladder epithelia during embryonic development. Our research detailed several features of the UCS epithelium according to generation number and embryonic stage. The differentiation scores of the epithelia reveal that the initial urothelial differentiation proceeds in a retrograde fashion in each region of the urinary tract. The present study provided histological data on the human urinary tract using embryos. Further study focusing on the histological development of the urinary tract in the fetus would be necessary to complete the chronology of UCS, ureter, and bladder histological development from conception to birth.

## Supporting information

**S1 Data.**
(XLSX)

**S2 Data.**
(XLSX)

**S3 Data.**
(XLSX)

## Acknowledgments

The authors thank Ms. Chigako Uwabe at the Congenital Anomaly Research Center for technical assistance in handling human embryos.

## Author Contributions

**Conceptualization:** Tetsuya Takakuwa.

**Data curation:** Marie Ange Saizonou, Haruka Kitazawa, Shigehito Yamada.

**Formal analysis:** Marie Ange Saizonou, Haruka Kitazawa.

**Methodology:** Toru Kanahashi.

**Supervision:** Tetsuya Takakuwa.

**Writing – original draft:** Marie Ange Saizonou.

**Writing – review & editing:** Tetsuya Takakuwa.

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
