## [Decision Letter · Decision Letter 0]

22 Feb 2024

PONE-D-23-31119Epithelial development of the urinary collecting system in the human embryoPLOS ONE

Dear Dr. Takakuwa,

Thank you for submitting your manuscript to PLOS ONE. After careful consideration, we feel that it has merit but does not fully meet PLOS ONE’s publication criteria as it currently stands. Therefore, we invite you to submit a revised version of the manuscript that addresses the points raised during the review process.

We look forward to receiving your revised manuscript.

Kind regards,

Antoine Naem, M.D.

Academic Editor

PLOS ONE

Journal Requirements:

Reviewers' comments:

Reviewer's Responses to Questions

**Comments to the Author**

1. Is the manuscript technically sound, and do the data support the conclusions?

Reviewer #1: Yes

Reviewer #2: Partly

2. Has the statistical analysis been performed appropriately and rigorously? 

Reviewer #1: N/A

Reviewer #2: Yes

3. Have the authors made all data underlying the findings in their manuscript fully available?

Reviewer #1: Yes

Reviewer #2: Yes

4. Is the manuscript presented in an intelligible fashion and written in standard English?

Reviewer #1: Yes

Reviewer #2: Yes

5. Review Comments to the Author

Reviewer #1: Authors have presented a well written study that describers histogenesis of the collecting system. Methods are sound and presentation of data is excellent. Diagrams are clear and well done.

Lines 97-98: would be better to clarify the criteria used for differentiation of the choroid plexus and what modifications were used

Line 462: 'retrogradely' is not grammatically correct, use 'in a retrograde fashion'

Line 464-466: Would read better as ' . (F)further study focusing on the histological development of

urinary tract in fetus would be necessary to complete the chronology of UCS, ureter and bladder histological developmentt from conception to birth.

Reviewer #2: The main point of this manuscript is that it describes an investigation of the UCS, ureter and bladder epithelia during early embryonic development. This would allow the authors to obtain better histological developmental data on the UCS, ureters, and bladder than had previously been available, and to use them to offer new possibilities for the clinical management of congenital urinary tract defects. This work seems worthwhile, but needs improvement to be published in PLOS ONE.

First, it is necessary to clearly state in the introduction why this study is important as a solution to the previous problems. Although this is a significant study, it is difficult to understand its purpose.

Second, there is no explanation of Figs 6 and 7 in the text, and Figs 8 and 9 are not sufficiently explained.

Furthermore, a detailed explanation is needed as to why the choroid plexus was chosen for this differentiation criterion.

6. PLOS authors have the option to publish the peer review history of their article (what does this mean?). If published, this will include your full peer review and any attached files.

Reviewer #1: No

Reviewer #2: No

---

## [Author Response · Author response to Decision Letter 0]

4 Mar 2024

Response to the Reviwers

Reviewer #1: 

1. Lines 97-98: would be better to clarify the criteria used for differentiation of the choroid plexus and what modifications were used

Explanations were added in Histological findings, Evaluation of the UCS epithelium, urinary duct, and bladder, in the Materials and Methods sections. 

Histological evaluation of the differentiation is not known for the human embryonic urothelium and is limited to another type of embryonic epithelium except for that of the choroid plexus [20,21]. The choroid plexus is a special organ that produces cerebrospinal fluid and is responsible for an important biological barrier system, forming an interface between the blood and the cerebrospinal fluid. Considering the urothelium's several functions described above, some physiological similarities in both structures appear evident. Therefore, it seemed interesting to use a similar differentiation criterion to describe epithelial tissue on the urinary tract whilst including its distinctive features.

2. Line 462: 'retrogradely' is not grammatically correct, use 'in a retrograde fashion'

The phrase in the conclusion was changed as indicated.

3. Line 464-466: Would read better as '. (F) further study focusing on the histological development of urinary tract in fetus would be necessary to complete the chronology of UCS, ureter and bladder histological development from conception to birth.

The sentence in the conclusion was changed as indicated. 

Reviewer #2: 

1. First, it is necessary to clearly state in the introduction why this study is important as a solution to the previous problems. Although this is a significant study, it is difficult to understand its purpose.

Fourth paragraph in Introduction was reworded for that the purpose and significance of this research become much clear.

Although the urinary tract (including the ureter and bladder) is uniformly lined by the urothelium, the process of histological differentiation during the embryonic period is primarily confined to the bladder [2,3,16] and urinary tract [6], with detailed descriptions lacking for the UCS [12]. Because urine secretion initiates during the late embryonic period [17], clarifying the features and timeline of the histological differentiation of the urothelium region by region is essential. Clinically, this could yield data to ascertain the locations of congenital urinary tract defects, potentially facilitating the initiation of a clinical retrospective study on probable causes. Herein, we aimed to demonstrate the differentiation of the UCS epithelium in the human metanephros during the human embryonic period and to evaluate its degree of differentiation compared to that of the ureter and bladder epithelia. 

2. Second, there is no explanation of Figs 6 and 7 in the text, and Figs 8 and 9 are not sufficiently explained.

Sufficient explanation of Figures 6, 7, 8, and 9 was added. Now reader could understand these Figures easily.

3. Furthermore, a detailed explanation is needed as to why the choroid plexus was chosen for this differentiation criterion.

Explanations were added in Histological findings, Evaluation of the UCS epithelium, urinary duct, and bladder, in the Materials and Methods sections. 

Histological evaluation of the differentiation is not known for the human embryonic urothelium and is limited to another type of embryonic epithelium except for that of the choroid plexus [20,21]. The choroid plexus is a special organ that produces cerebrospinal fluid and is responsible for an important biological barrier system, forming an interface between the blood and the cerebrospinal fluid. Considering the urothelium's several functions described above, some physiological similarities in both structures appear evident. Therefore, it seemed interesting to use a similar differentiation criterion to describe epithelial tissue on the urinary tract whilst including its distinctive features.

---

## [Decision Letter · Decision Letter 1]

22 Mar 2024

Epithelial development of the urinary collecting system in the human embryo

PONE-D-23-31119R1

Dear Dr. Takakuwa,

We’re pleased to inform you that your manuscript has been judged scientifically suitable for publication and will be formally accepted for publication once it meets all outstanding technical requirements.

Kind regards,

Antoine Naem, M.D.

Academic Editor

PLOS ONE

Reviewers' comments:

Reviewer's Responses to Questions

**Comments to the Author**

1. If the authors have adequately addressed your comments raised in a previous round of review and you feel that this manuscript is now acceptable for publication, you may indicate that here to bypass the “Comments to the Author” section, enter your conflict of interest statement in the “Confidential to Editor” section, and submit your "Accept" recommendation.

Reviewer #1: All comments have been addressed

Reviewer #2: All comments have been addressed

2. Is the manuscript technically sound, and do the data support the conclusions?

Reviewer #1: Yes

Reviewer #2: Yes

3. Has the statistical analysis been performed appropriately and rigorously? 

Reviewer #1: Yes

Reviewer #2: N/A

4. Have the authors made all data underlying the findings in their manuscript fully available?

Reviewer #1: Yes

Reviewer #2: Yes

5. Is the manuscript presented in an intelligible fashion and written in standard English?

Reviewer #1: Yes

Reviewer #2: Yes

6. Review Comments to the Author

Reviewer #1: All questions are well addressed. Well done to the authors. I have no further suggestions or corrections.

Reviewer #2: The authors have responded to each of my points and they have been careful simply to describe what happened in the USC development. I would support the publication of this amended version in PLOS ONE.

7. PLOS authors have the option to publish the peer review history of their article (what does this mean?). If published, this will include your full peer review and any attached files.

Reviewer #1: No

Reviewer #2: No

---

## [Editor Report · Acceptance letter]

30 Mar 2024

PONE-D-23-31119R1 

PLOS ONE

Dear Dr. Takakuwa, 

I'm pleased to inform you that your manuscript has been deemed suitable for publication in PLOS ONE. Congratulations! Your manuscript is now being handed over to our production team.

Kind regards, 

on behalf of

Dr. Antoine Naem 

Academic Editor

PLOS ONE